# Co-occurrence patterns of malnutrition indicators among children in sub-Saharan Africa
Johannes Seiler [1] ✉, Benjamin Müller [1], Isabel Günther[2,3], Mattias Wetscher [1], Reto Stauffer [1,4], Nikolaus Umlauf [1] & Kenneth Harttgen [2,3] ✉

## Abstract

**Background** Despite progress towards SDG2, malnutrition remains a major concern. Little is known about co-occurrence patterns of nutritional deficits, particularly in low- and middle-income countries. This study analyzes four key child nutrition indicators: hemoglobin (Hb), height-for-age z-score (HAZ), weight-for-age z-score (WAZ), and weight-for-height z-score (WHZ), to identify shared risk factors and spatio-temporal dynamics of malnutrition in sub-Saharan Africa (SSA).

**Methods** Data on 205,374 children aged six to 59 months, of both sexes, from 30 countries in SSA, surveyed between 2003 and 2020, were obtained from the Demographic and Health Surveys. These data were merged with geospatial information from various sources. The generalized additive models for location, scale and shape framework was used to fit a multivariate Gaussian model to investigate co-occurrence patterns, shared risk factors and spatio-temporal variability of malnutrition indicators in SSA.

**Results** The analysis reveals substantial variability in the analyzed indicators across age groups and geographic regions. In 2020, among included countries, the corresponding estimated prevalence rates were: (i) anemia: 69.6% [66.2% -72.8%]; (ii) stunting: 32.9% [32.2% -33.6%]; (iii) underweight: 14.8% [13.6% -15.9%]; (iv) wasting: 5.2% [4.5% -6.0%]; and (v) overweight: 2.5% [2.0% -3.1%]. Strong correlations were observed between WAZ and WHZ (0.70 [0.67-0.73]) and HAZ and WAZ (0.68 [0.64-0.71]), suggesting that chronic malnutrition often co-occurs with acute malnutrition. In contrast, the correlation between anthropometric indicators and the Hb level was relatively low, although geographic variability still highlights specific hotspots.

**Conclusions** We provide high-resolution prevalence estimates for anemia, stunting, underweight, wasting, and overweight, alongside their pairwise correlations. The pronounced spatial heterogeneity highlights the need for localized, coordinated interventions. The findings support multi-sectoral strategies involving nutrition, health, education, and social protection programs to reduce malnutrition in SSA.

## Plain Language Summary

To better understand how different forms of malnutrition occur together in sub-Saharan Africa, we jointly analyzed four key measures of malnutrition that are commonly used to identify anemia, short-term and long-term undernutrition, as well as overnutrition, in children between the ages of six and 59 months. Machine learning methods were used to identify shared risk factors and to uncover patterns across space and time associated with these conditions. Our findings reveal strong interrelations between the measures of malnutrition, which show substantial variability across age groups. This highlights the fact that chronic malnutrition frequently overlaps with acute malnutrition and anemia. These findings underscore the need for more targeted public health interventions to address these pressing issues.

Over the past three decades, significant efforts have been made to reduce malnutrition[1]. Despite these efforts, the risk of both child mortality and malnutrition remains substantially higher in low- and middle-income countries (LMICs) than in high-income countries. Globally, millions of children under the age of five continue to suffer from malnutrition, typically measured by anthropometric indicators such as stunting, wasting, underweight status, or overweight status. In addition, they often manifest in medical conditions such as anemia, with the vast majority of affected populations living in South Asia and sub-Saharan Africa (SSA)[1,2].

[1]Department of Statistics, University of Innsbruck, Innsbruck, Austria. [2]Development Economics Group, ETH Zurich, Zurich, Switzerland. [3]NADEL Center for Development and Cooperation, ETH Zurich, Zurich, Switzerland. [4]Digital Science Center, University of Innsbruck, Innsbruck, Austria. ✉e-mail: Johannes.Seiler@uibk.ac.at; kenneth.harttgen@nadel.ethz.ch

In 2022, approximately 31.3% of children under the age of five in SSA were chronically undernourished (i.e., stunted), a condition characterized by significantly reduced height-for-age due to factors such as prolonged poor nutrition and recurrent infections. This translates to around 56.8 million stunted children in SSA. Additionally, around 5.7%–or approximately 10.3 million–of children under the age of five in SSA were suffering from acute undernourishment (i.e., wasting). This condition is characterized by a low weight-for-height ratio, often associated with inadequate nutritional intake and frequent illness[1]. Moreover, around 3.7%–or approximately 6.6 million–of children under the age of five in SSA were overweight, a condition defined by a weight-for-height ratio that is two standard deviations above the World Health Organization (WHO) growth standard[1,3]. Overweight is often associated with poor availability and access to nutritious foods, leading to the substitution of nutrient-dense foods for calorie-dense, poor alternatives[1]. Additionally, in 2019, approximately 61.0% of children between six and 59 months of age in SSA–equivalent to around 102.3 million children–were affected by anemia, a condition caused by insufficient intake of micronutrients (i.e., hidden hunger), infections with parasites, or infectious diseases, with iron deficiency and malaria being the most common causative factors[4–9]. Anemia is characterized by a reduced number of erythrocytes or hemoglobin (Hb) levels below a predefined cutoff[10–13]. These persistent issues highlight the ongoing public health crisis in LMICs and particularly in SSA, where the commitments made under the 2030 Agenda for Sustainable Development and particularly Sustainable Development Goal (*SDG*) 2–aiming to end hunger, achieve food security, and improve nutrition–are far from being met[2,9,14].

The observed patterns of malnutrition are closely linked to child mortality, as malnutrition is considered a major risk factor for child mortality[15–19]. It is estimated that approximately 35% to 45% of all child deaths globally among children under five years–nearly half of all child deaths–can be attributed to some form of malnutrition[17,18,20,21]. Among other factors, malnutrition weakens the immune system and impairs growth and development, making anemia and undernutrition major contributors to infant and under-five mortality. For instance, anemia, particularly due to iron and vitamin deficiency, further increases children's vulnerability and increases the risk of infections and complications, which can also lead to a higher risk of early child death. Despite substantial global progress, infant and under-five mortality rates remain high in some regions, particularly in SSA, where the estimated infant (under-five) mortality rate was 4.4% (6.8%) in 2023. These rates are substantially higher than the reported infant (under-five) mortality rate in South Asia, the second most affected region, where the rate was 3.0% (4.5%), and about nine times higher than in North America in the same year. In absolute terms, about two million children under the age of one died in SSA, accounting for about 50% of all global child deaths in 2023[19]. Understanding and effectively addressing the nutritional aspects of early childhood development is therefore not only a matter of meeting *SDG* 2, but also critical to *SDG* 3, which aims to ensure healthy lives and promote well-being for all at all ages, including specific targets to reduce neonatal and under-five mortality in all countries by 2030.

Recent studies have increasingly used joint and multivariate modeling techniques to explore co-occurrence patterns of child malnutrition indicators across several countries in SSA. For instance, a multilevel multivariate analysis based on Demographic and Health Survey (DHS) data from ten East African countries revealed significant interrelations among stunting, wasting, and underweight, with child age, maternal education, and access to improved drinking water emerging as key covariates[22]. In Tanzania, spatial analyses focusing on young children revealed considerable geographic overlap between stunting, wasting, and underweight, emphasizing the influence of maternal factors and regional variability[23,24]. Moreover, joint spatial mapping efforts across SSA have highlighted localized disparities in nutritional outcomes, underscoring the necessity of geographically targeted nutrition interventions[25].

Building on this body of research, a flexible multivariate Gaussian regression framework has been developed to jointly model the four key measures of malnutrition: Hb, height-for-age z-score (HAZ), weight-for-

age z-score (WAZ), and weight-for-height z-score (WHZ). These indicators are commonly used to assess various forms of malnutrition, including anemia, stunting, underweight status, overweight status, and wasting. This approach allows to incorporate nonlinear covariate effects, automatic variable selection, and high-resolution spatial prediction across many countries in SSA. By simultaneously capturing co-occurrence patterns and shared risk factors, this framework provides high-resolution estimates of both prevalence levels and pairwise correlations. This facilitates a more comprehensive understanding that different indicators of malnutrition are interrelated across demographic and geographic dimensions.

This study addresses a critical gap in malnutrition research by jointly analyzing key nutritional indicators (i.e., Hb, HAZ, WAZ, and WHZ) using individual-level data from 30 countries in SSA over multiple years. Unlike previous work focused on single-indicator, single-country, or single-year datasets[23,24,26–29], this multi-country, multi-year approach offers a broader perspective and enables examination of co-occurrence patterns largely overlooked in existing literature. Using supervised statistical learning methods, this study contributes to the literature in the following ways: (i) Examine co-occurrence patterns, prevalence levels, and trends of multiple malnutrition indicators; (ii) Identify shared risk factors, including individual, socio-economic (e.g., gender, maternal education, household wealth, healthcare and sanitation access), and environmental factors (e.g., water availability, infectious disease exposure, climatic conditions); and (iii) Analyze spatio-temporal dynamics, producing high-resolution (20 × 20 km) estimates of pairwise correlations. The study also maps geographic variations to identify hotspots, providing evidence to guide targeted and coordinated public health interventions across SSA.

Analyzing the pairwise correlation between the nutritional indicators can be of particular relevance for policymakers and non-governmental organizations (NGOs), as it can help to identify high-risk areas and particularly vulnerable age groups of children who are affected by multiple forms of malnutrition. In particular, combinations of high positive pairwise correlations and, for instance, low levels in the considered health indicators are of great concern, as they imply that co-occurrence and the resulting negative consequences are widespread among children, highlighting a need for urgent action to address undernutrition and hidden hunger. In addition, joint modeling of these indicators will help to provide information for targeted interventions, will help policymakers to design efficient public health strategies, and support the monitoring of *SDG* 2 and *SDG* 3 in SSA.

Our analysis reveals that the indicators under study exhibit substantial variability across both age groups and geographic regions. Additionally, we observe strong correlations between these indicators, suggesting that chronic malnutrition frequently overlaps with acute malnutrition and anemia in specific regions. These findings emphasize the need for better-targeted combined public health interventions to address malnutrition, particularly in high-prevalence areas.

## Methods

### Measurement of undernutrition in children

The United Nations Children's Fund (UNICEF) conceptual framework on maternal and child nutrition, developed in 1990 and refined over the years[17,18,30,31], provides a comprehensive approach to understanding child malnutrition. It goes beyond physiological aspects to include household, socio-economic and environmental dimensions. The framework identifies three levels of determinants: immediate determinants (such as dietary intake and health status), underlying determinants (such as food security, maternal care, and access to health services), and basic (enabling) determinants (including economic and social conditions at the regional or national level, and environmental and climatic conditions). This broader perspective is essential to capture the multiple factors that influence child nutrition and to guide interventions that target both the immediate and underlying causes of malnutrition[32].

Based on this framework, child malnutrition is typically assessed using anthropometric indicators that reflect both long-term and short-term nutritional deficiencies. Stunting (i.e., HAZ<−2) and underweight status (i.e.,

WAZ<−2) are used to assess chronic malnutrition, indicating long-term growth deficiencies[33,34]. Wasting (i.e., WHZ<−2), along with the underweight status and other indicators such as mid-upper arm circumference, are used to assess acute malnutrition[33,35]. These indicators are critical for understanding child growth and development, with stunting being particularly indicative of long-term nutritional deficits[36]. In addition to stunting, underweight status, and wasting, which are indicators of different forms of malnutrition, hidden hunger refers to micronutrient deficiencies that may not be visible but still affect children's health and development. These deficiencies often go undetected by standard anthropometric measures, but are critical for assessing the overall malnutrition status[37]. One condition that partly reflects micronutrient deficiencies is anemia. Anemia is characterized by the Hb level in the blood of an individual below a pre-defined age-, elevation-, and sex-specific threshold[10,11]. According to the WHO, the cutoff to define anemia in children between 6 and 59 months is a Hb level below 110 gL$^{-1}$[5,12]. Given these considerations, the four malnutrition indicators briefly outlined below, and in *Supplementary Note*2 (see *Supplementary Table*3), can be analyzed together to investigate potential co-occurrence patterns.

## Statistics and reproducibility
The following sections provide a concise overview of the modeling approach used in this study. This overview guides the interpretation of the results and ensures the reproducibility of the statistical analysis. All methodological details, including model specification and implementation, are outlined in *Supplementary Note*2.

## Data
This study aims to bridge the gap and enhance the understanding of potential co-occurrence patterns of commonly used indicators of malnutrition in SSA by merging DHS data[38] from 30 countries in SSA between 2003 and 2020 with data on climate, demographic, and environmental factors. Therefore, we analyze four routinely collected indicators–Hb, HAZ, WAZ, and WHZ–that are commonly used to assess anemia, and others forms of malnutrition, in children between six and 59 months of age (routine surveys, such as the DHS, do not measure Hb levels in children under 6 months of age because this could lead to biased estimates of anemia prevalence[39]).

Analyzing extensive household survey data, such as several distinct DHS data sets[38] at a disaggregated level, and the merging of these surveys with remotely sensed or spatial information has proven particularly useful for monitoring the progress toward the SDGs. Potential useful spatial covariates are, for example, land-cover[40–42], precipitation[43], soil type[44], or temperature[43]. Examples using geo-spatial information include e.g., refs. 45–50, which analyze the spatio-temporal dynamics and progress of various metrics related to the SDGs.

By merging DHS data with demographic, environmental, and geospatial factors, a unique data set has been created that leverages shared characteristics across countries, space, and time and can be used to identify socio-economic, environmental, and spatial factors that are associated with different indicators of malnutrition. This allows for a comprehensive analysis of *SDG* 2-relevant indicators–Hb, HAZ, WAZ, and WHZ–using an approach for multivariate Gaussian regression that is based on a (modified) Cholesky decomposition[51] and is embedded in the framework of generalized additive models for location, scale, and shape (GAMLSS;[52]), also known as distributional regression[53–55]. An in-depth discussion of the analytical framework of distributional regression, and on the included covariates is provided in several different applications, see e.g., refs. 50,56. In addition, see *Supplementary Table*1 and *Supplementary Figs.*1 and 2 for detailed information on the data sources, the pre-processing steps, and the geographic coverage of the analysis.

## Demographic and health surveys
The DHS[38] program forms the core data set used to derive the results of this study. The DHS are large household surveys conducted in LMICs that started in 1984 as the successor to the World Fertility Survey. They provide representative data on population, health, infectious diseases, and nutrition

for more than 90 countries. DHS routinely collects data on anthropometric indicators, as well as biomarkers, such as hemoglobin levels. Additionally, DHS surveys adhere to WHO guidelines for measuring both anthropometric indicators and biomarkers. Because of their geo-referenced nature, it is possible to merge DHS data with environmental or remotely sensed information from other sources.

## Merging DHS data sets with environmental and climate data
From the child recode of the DHS data[38], the indicators to determine the nutritional status as well as socio-economic covariates are obtained. These cross-sectional data are representative at the first administrative level and include, in addition to information on the response variables, information on, for example, the socio-economic background of the household to which a child belongs. See, for example, Corsi et al.[57] for more detailed information on the DHS data sets. One of the goals of this study is to examine potential spatio-temporal patterns in the co-occurrence of different forms of malnutrition. To allow spatial extrapolation to locations without direct ground measurements of these conditions, the DHS survey data is merged with socio-economic, environmental, and climatic covariates from various sources, such as SoilGrids[44]–global gridded soil information, or European Center for Medium-Range Weather Forecasts (ECMWF) ERA5[43]–global climate and weather reanalysis. See *Supplementary Fig.*2 for information on the spatial coverage of this analysis, and Seiler et al.[50], *Supplementary Table*1, and *Supplementary Fig.*1 for further discussion and information on data used in this study.

## Modeling approach
This study employs a multivariate modeling approach similar to the univariate approach described by Seiler et al.[50] for modeling anemia prevalence in SSA and South Asia. Further methodological considerations can be found in Umlauf and Kneib[58]. The five distinct steps are: (i) *Creating the training and test data set*. The data is split into an 80% training data set and a 20% test data set. Delaunay triangulation is used to ensure spatial representativeness in SSA countries[59], as opposed to random splitting. (ii) *Estimating a baseline reference model*. An initial model is estimated without covariates (i.e., intercept-only model), using the mean vector $\boldsymbol{\mu}$ and variance-covariance matrix $\boldsymbol{\Sigma}$. This model serves as baseline model used for comparison purposes. (iii) *Identifying informative covariates in each predictor$\eta_k$*. Informative covariates for the multivariate distributional regression model are selected using a boosting algorithm[60]. (iv) *Bayesian estimation of the final model omitting uninformative covariates*. The final model is estimated using Markov chain Monte Carlo (MCMC) sampling based on all informative covariates that were identified in the previous step. (v) *Model validation*. Model calibration and predictive performance of the final model are assessed using graphical methods, different validation metrics, and a cross-validation routine to investigate potential data sparsity issues. The following provides an overview of the GAMLSS framework for multivariate response vectors. Further details are provided in *Supplementary Note*2 and *Supplementary Note*3.

## Multivariate Gaussian regression
Multivariate Gaussian regression is integrated into the GAMLSS framework[52]. The methodological flexibility, combined with the ease of interpretation, and the strong predictive performance, has made GAMLSS a valuable tool in many univariate applications related to Millennium Development Goals (MDG), as well as SDG progress monitoring (e.g., refs. 45–47,50,56,61). However, extending the scope to the analysis of multivariate responses–and thus studying co-occurrence patterns and shared risk factors–has been less common in practice. Notable exceptions include studies by Klein et al.[55,62], which jointly analyze acute and chronic undernutrition in India.

Multivariate Gaussian regression extends the scope from a univariate response $y_i$, with $i = 1, \ldots, n$ observations (e.g., the modeling of the HAZ) to the analysis of an $M$-dimensional multivariate Gaussian response vector $\mathbf{y}_i = (y_{i1}, \ldots, y_{iM})^{\top}$, i.e., $\mathbf{y}_i = (\text{Hb}_i, \text{HAZ}_i, \text{WAZ}_i, \text{WHZ}_i)^{\top} \sim \mathcal{N}(\boldsymbol{\mu}_i, \boldsymbol{\Sigma}_i)$. Here, $\boldsymbol{\mu}_i$ corresponds to the mean vector and $\boldsymbol{\Sigma}_i$ to the variance-covariance

**Fig. 1 | Barplot of the frequencies of selected model terms.** The color shading represents the distributional parameters ($\mu$, $\sigma$, and $\rho$) associated with the selected term. Further, note that in the context of the GAMLSS framework, the idea is to characterize all distributional parameters of an arbitrary (multivariate) response distribution using covariates. Here $\mu$ corresponds to the location parameters (i.e., means), $\sigma$ to the scale parameters (i.e., variances), and $\rho$ to the correlation parameters (i.e., pairwise correlations). See *Supplementary Table* 3 for details on the used covariate abbreviations. Note that these results are based on the final model, which was fitted using the training data ($n$=164,300) and validated using the test data ($n$ = 41,074). For more details, see *Supplementary Fig.* 1.

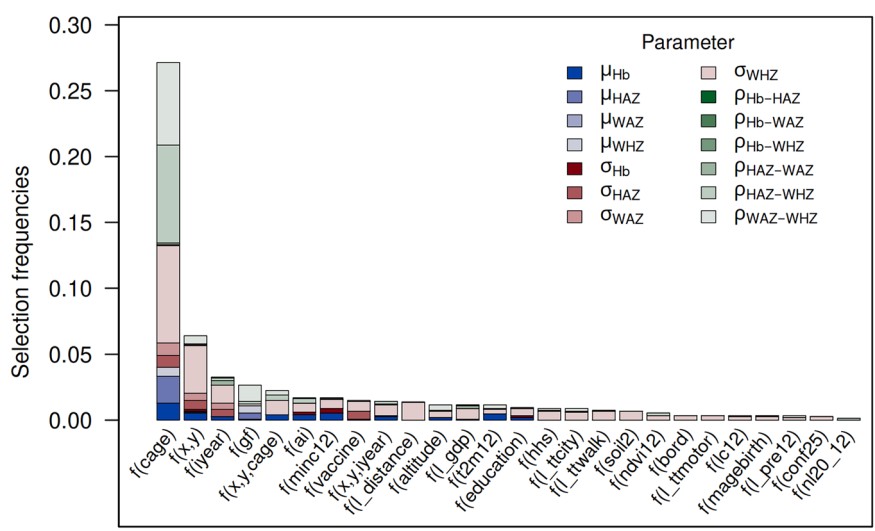

matrix of the four metric outcomes of interest (Hb, HAZ, WAZ, and WHZ) that are commonly used to measure different forms of malnutrition and hidden hunger. Accordingly, the response vector **y** is expressed by a four-dimensional multivariate Gaussian distribution. The variance-covariance matrix $\Sigma$ is a $4 \times 4$ matrix, and $\Sigma$ contains $4 \cdot (4 + 1)/2 = 10$ unique entries. Therefore, in total, this four-dimensional multivariate Gaussian regression model consists of 14 distributional parameters (4 location parameters, 4 scale parameters, and 6 correlation parameters) that are modeled as functions of the covariates. To estimate the model, the (modified) Cholesky decomposition proposed by Pourahmadi[63] and implemented by Muschinski et al.[51] is used. This approach not only ensures positive definiteness of the variance-covariance matrix, but also allows that all parameters–not only $\boldsymbol{\mu}_i$ (the mean vector)–of the multivariate Gaussian distribution can be related to additive predictors. Accordingly, all parameters of the response distribution can be modeled using covariate information, such as socio-economic, environmental, or climate factors related to the indicators studied here. This approach allows for the analysis of co-occurrence patterns and shared risk factors.

## Ethical approval

This study is based on secondary analyses of publicly available Demographic and Health Surveys (DHS) data. The data are fully de-identified and were combined only with non-human data sources, hence, no additional ethical approval was required. Access to the DHS data sets is granted upon registration through the DHS data repository[38]. Data collection and dissemination adhere to established ethical standards for research involving human subjects, as described in the *Protecting the Privacy of DHS Survey Respondents* documentation[64]. All DHS survey protocols have been reviewed and approved by ICF Institutional Review Board (IRB), as well as by relevant ethical review boards in the respective host countries. Informed consent is obtained from all respondents, and for children, consent is provided by a parent or legal guardian prior to participation. Consequently, further IRB approval is not required for research relying solely on DHS data. None of the additional data sources used in this study contain human subjects data.

## Results
### Selected covariates

Applying the batchwise backfitting algorithm introduced by Umlauf at al.[65], allows for identifying the most informative predictors associated with the measures of malnutrition. Based on this approach (see *Supplementary Note* 2.2), the covariates found to be most relevant are, the age of the child and the location of the primary sampling unit where the child lives. See Fig. 1

for more details and all covariate terms selected by this approach. The frequent selection of the children's age and the geographic location across all distributional parameters highlights two important findings: Firstly, the analyzed indicators of malnutrition are not evenly distributed across the age range of 6 to 59 months. In contrast, the estimates suggest a more nuanced picture, with younger children having a higher risk of being anemic, while older children have a higher risk of being acutely and chronically malnourished. Secondly, in areas experiencing chronic malnutrition (as indicated by stunting) and a positive correlation between HAZ and other malnutrition indicators, acute malnutrition and anemia often accompany stunting. This underscores the necessity of implementing targeted interventions that simultaneously address hidden hunger, acute malnutrition, and chronic malnutrition in these regions. Other covariates that are selected at a much lower frequency include the interview year, the child's sex, the household wealth index, and socio-economic and environmental characteristics such as surface temperature, immunization status, gross domestic product, and malaria incidence. This underlines two further important aspects: (i) the importance of household characteristics, and (ii) the influence of environmental factors on a child's nutritional status.

### Estimated prevalence and trends of anemia, stunting, underweight, wasting, and overweight

Figure 2 (see *Supplementary Figs.* 6 to 8 for the corresponding estimates including uncertainty intervals) presents model-based estimates of the spatio-temporal dynamics of anemia, stunting, underweight, wasting, and overweight among children aged six to 59 months, disaggregated at the pixel level ($20 \times 20$ km) across SSA for the years 2010, 2015, and 2020. These high-resolution estimates reveal distinct spatial and temporal patterns for each indicator. Three key observations emerge: (i) consistent with other data sources[6,7,46,48,50,66], the prevalence of all five indicators declined over time, although at varying rates. Notably, the prevalence of anemia and overweight showed only modest improvements, with some countries even experiencing increases; (ii) the prevalence of anemia, stunting, and underweight remains high in many SSA countries, exceeding the WHO thresholds for serious public health concern of 40%, 30%, and 20%, respectively; (iii) the prevalence of overweight is estimated to be 2.5% [95% credible interval: 2.0–3.1%] in 2020; and (iv) the Sahel region in western SSA consistently shows high prevalence across all five indicators. These findings are consistent with previous studies[46,48,50] and national-level estimates from UNICEF and WHO[7,66]. Beyond these overarching trends, distinct hotspots and coldspots can also be identified throughout SSA.

**Fig. 2 | Estimated prevalence of anemia, stunting, underweight, wasting, and overweight among children aged 6 to 59 months in SSA in 2010, 2015, 2020. a–c** anemia (i.e., $P$(Hb<110 $g\,L^{-1}$)); **d–f** stunting (i.e., $P$(HAZ<−2); **g–i** underweight (i.e., $P$(WAZ<−2); **j–l** wasting (i.e., $P$(WHZ<−2); and **m–o** overweight (i.e., $P$(WAZ>2). Black lines reflect country-level administrative borders. Pixels categorized as *Barren*, or *Permanent Snow and Ice*, and pixels above 3000 m are flagged as *Not included*. Note that these predictions are based on the final model, which was fitted using the training data ($n = 164,300$) and validated using the test data ($n = 41,074$). For more details, see *Supplementary Fig.*1.

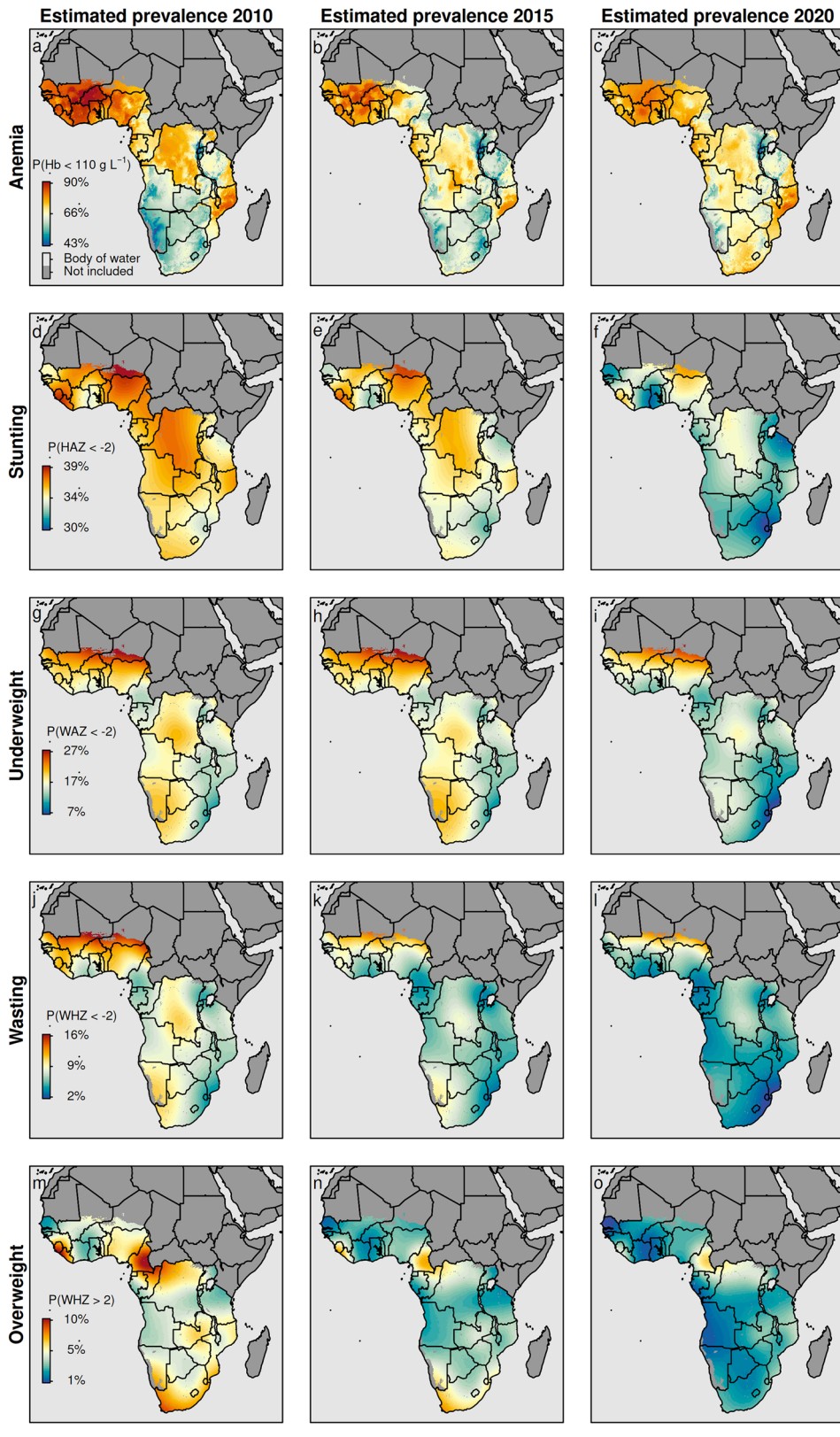

In 2020, the highest prevalence of anemia among children aged 6 to 59 months was observed in Burkina Faso, at 80.0% [78.1–81.9%], followed by Côte d'Ivoire (79.1% [77.1–81.1%]) and Mali (78.7% [76.7–80.6%]), all located within or bordering the Sahel zone. Despite a decrease of approximately five to eight percentage points compared to 2010 estimates, anemia levels in these countries remain extremely high. Conversely, the lowest

anemia prevalence was found in Rwanda (48.5% [46.6–50.3%]), Burundi (54.5% [52.7–56.4%]), and Uganda (61.1% [58.5–63.8%]), all in the eastern part of SSA. However, even these values exceed the WHO threshold of 40%, indicating a severe public health concern.

Stunting prevalence was highest in Niger (35.8% [35.2–36.4%]), followed by Sierra Leone (35.2% [34.7–35.7%]), and Liberia (34.9%

[34.4–35.5%]), all located in western SSA. Compared to 2010, these countries showed a modest decline of around five percentage points. Underweight prevalence was led by Niger (22.8% [21.8–23.9%]), followed by Mali (20.2% [19.5–20.9%]), and Burkina Faso (18.9% [18.2–19.6%]), with only slight reductions since 2010. Wasting prevalence was highest in Niger (10.8% [9.8–11.8%]), Mali (9.2% [8.6–9.9%]), and Burkina Faso (8.0% [7.4–8.6%]), also showing minimal improvements over the past decade. In contrast, the highest prevalence of overweight in 2020 was recorded in Cameroon (5.1% [4.6–5.6%]), followed by Sierra Leone (4.0% [3.7–4.3%]) and Rwanda (3.3% [3.0–3.7%]).

However, the individual prevalence estimates are generally more accurate in a univariate modeling setting due to the more flexible choice of distributions and the ability to fully tailor spatial components to each indicator. In contrast, the multivariate model does not select all spatial terms for every outcome, potentially affecting marginal prevalence estimates. Therefore, we restrict our discussion to country-level results and emphasize that the estimates reported here should be considered baseline approximations. These prevalence estimates primarily serve a contextual background, as our primary interest lies in understanding co-occurrence patterns.

*Supplementary Figs.*9 to 11 provide high-resolution estimates of anemia, stunting, underweight, wasting, and overweight disaggregated by age in 18-month intervals. While the risk of anemia and the prevalence of wasting and underweight are highest among younger children–particularly those aged 6 to 24 months–the prevalence of stunting increases with age, peaking between the second and third years of life. It then persists at high levels until the age of five.

This contrast reflects the different etiologies of these conditions: wasting and anemia are typically acute conditions, often peaking during the weaning period when children are exposed to infections and inadequate complementary feeding. In contrast, stunting and underweight represent chronic undernutrition that accumulates over time due to prolonged dietary deficiency and repeated illness.

These patterns underscore the importance of the first 1,000 days of life–including pregnancy and the first 2 years postpartum–as a critical window for intervention. In particular, preventing wasting and anemia during infancy is essential to reduce child mortality and support healthy development, while monitoring for growth faltering and excess weight gain in older children remains equally vital.

### Patterns of co-occurrence

Based on the compiled DHS data[38], the pairwise correlation between the four responses of interest ranges from about −0.10 (between HAZ and WHZ) to about 0.68 between WAZ and WHZ, which is partly due to the way the z-scores are constructed. Furthermore, when looking at the pairwise correlations at the country-level and over time, a similar but more nuanced picture emerges, where the variation in the correlation coefficients between countries outweighs the temporal variation. See *Supplementary Table*2 for an overview of the calculated pairwise correlation coefficients at the country-year level.

Figure 3 shows the estimated pairwise correlation between the included measures of malnutrition within SSA for children between 6 and 59 months of age for the year 2010 and 2020. See *Supplementary Figs.*12 to 14 in *Supplementary Note*5 for more detailed results on the estimated pairwise correlation for children between 6 and 23 months of age, between 24 and 41 months of age, and between 42 and 59 months of age, respectively. Note that in the upper triangular panel, the estimated pairwise correlation coefficients are shown for the year 2020, while the lower triangular panel shows the estimated pairwise correlation coefficients for the year 2010. Two aspects are striking: the correlation between the Hb level, which determines the anemia status, and the anthropometric indicators is rather low, ranging from a pairwise correlation coefficient of −0.01 to a pairwise correlation coefficient of about 0.18. See the corresponding panels of Fig. 3, as well as *Supplementary Figs.*12 to 14 in *Supplementary Note*5, which show the magnitude of the estimated correlation coefficient at a given location over

time. To some extent contrary to this, the estimated pairwise coefficient of correlation of the three different anthropometric indicators (i.e., HAZ, WAZ, and WHZ) reaches up to about 0.8. Surprisingly, based on the model estimates the coefficient of correlation between the anthropometric indicator used to identify acute undernutrition (i.e., WHZ) and chronic undernutrition (i.e., HAZ) shows a negative correlation. A similar finding was also highlighted by Klein et al.[55] for the bivariate analysis of acute and chronic undernutrition in India.

The estimated correlation between Hb level and anthropometric scores is estimated to be highest in Uganda, Ghana, Cote d'Ivoire, and Mozambique. These countries are also among the LMICs with a high prevalence (see, for example,[46,48,50] for the corresponding high-resolution estimates).

### Shared risk factors

Anemia and wasting are most prevalent in young children under the age of two, and their prevalence decreases non-linearly with age (see Fig. 4 and *Supplementary Figs.*9 to 11). In contrast, stunting also shows a nonlinear trend (see Fig. 4 and *Supplementary Figs.*9 to 11). However, it is least common in younger children, but progressively increases with age, leveling off around the third year. This suggests that the immediate effects of undernutrition, such as anemia and wasting, are more prominent in younger children and can often be linked to acute deficiencies of nutrients or repeated infections[67,68]. As children age, the long-term consequences of malnutrition or repeated infection, particularly stunting, become more evident, reflecting the cumulative impact of insufficient nutrition over time. These findings are consistent with those of Klein et al.[55], who documented similar trends between wasting and stunting in India.

The observed patterns underscore the need for combined, targeted, and comprehensive interventions–such as anti-malarial bed net campaigns, nutritional fortification, growth monitoring, and water, sanitation, and hygiene (WaSH) programs–that address the underlying causes of health disparities in vulnerable populations. Previous studies[69,70] have highlighted the need for such interventions.

Further analysis of the prevalence of anemia, stunting, underweight, and wasting, alongside the pairwise correlations (see Figs. 4 and 5 for age-stratified prevalence and correlation estimates), supports these patterns. Specifically, anemia, underweight, and wasting are most prevalent among younger children, with a clear decline as they age. In contrast, stunting–an indicator of chronic malnutrition–is less common in younger children but steadily increases with age, eventually plateauing in older groups. This highlights the dynamic relationship between age, nutritional status, and overall health. It further suggests that children born within the biologically normal height range may initially experience acute forms of malnutrition. This malnutrition manifests as underweight, wasting, and anemia due to insufficient nutrient intake and higher exposure to infections and parasites. Over time, these acute conditions contribute to the longer-term manifestation of stunted growth, a more persistent form of malnutrition.

However, these patterns are influenced by various covariates, including factors related to malaria prevalence. Environmental and socio-economic conditions–such as the location, elevation, malaria incidence, and household wealth–can significantly affect health outcomes. For instance, higher-altitude regions may have lower malaria transmission rates, which could lead to different health patterns compared to areas where malaria is more prevalent. Additionally, households with limited resources may have restricted access to malaria prevention measures, like insecticide-treated bed nets, or may face challenges in obtaining timely medical treatment. These factors could contribute to variations in both malnutrition and overall health status, as malaria can interfere with nutrient absorption and exacerbate existing health issues.

### Discussion

Before discussing the scope and implications of our findings, it is crucial to acknowledge that the results we present are descriptive. This study was not designed to establish causal relationships between covariates and the co-occurrence of anemia and different types of malnutrition. Rather, the study

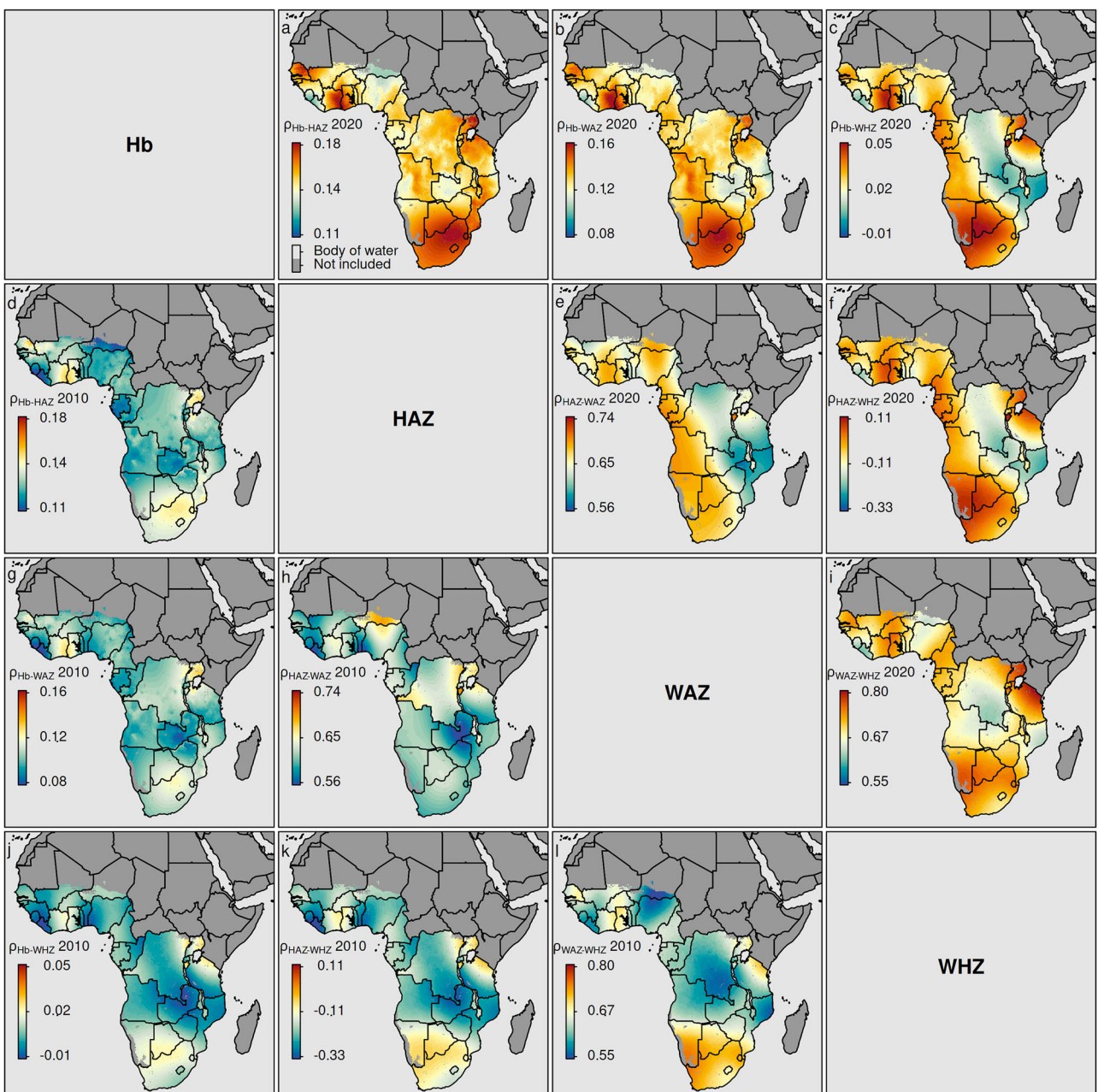

**Fig. 3 | Estimated pairwise correlation between four key nutritional indicators among children aged six to 59 months in SSA. a** Hb level and HAZ, 2020; **b** Hb level and WAZ, 2020; **c** Hb level and WHZ, 2020; **d** Hb level and HAZ, 2010; **e** HAZ and WAZ, 2020; **f** HAZ and WHZ, 2020; **g** Hb level and WAZ, 2010; **h** HAZ and WAZ, 2010; **i** WAZ and WHZ, 2020; **j** Hb level and WHZ, 2010; **k** HAZ level and WHZ, 2010; and **l** WAZ and WHZ, 2010. Black lines reflect country-level administrative borders. Pixels categorized as *Barren*, or *Permanent Snow and Ice*, and pixels above 3000 m are flagged as *Not included*. Note that these predictions are based on the final model, which was fitted using the training data (*n* = 164,300) and validated using the test data (*n* = 41,074). For more details, see *Supplementary Fig.* 1.

identifies geographic hotspots and coldspots where multiple nutrition-related indicators are poor or favorable simultaneously, providing valuable insights into spatio-temporal patterns and relationships.

It is important to consider certain limitations when interpreting the results: (i) the uncertainty in the estimates is particularly high in areas where data is sparse. This is the case in regions that are not densely populated (e.g., desert-like areas in Burkina Faso, Namibia, Niger, and South Africa) or where the sample size is small, highlighting the need for more recent and updated survey statistics. Additionally, the four selected indicators for joint analysis were not consistently observed for all eligible individuals, reducing the overall sample size. Consequently, certain countries, such as Ethiopia and Kenya, had to be excluded due to either geographical disconnection or

incomplete covariate information. (ii) While great care is taken in the pre-processing steps to merge data from different sources, some input data are pre-existing and modeled estimates (e.g., malaria prevalence). These esti-mates can be subject to inaccuracies that may be reflected in our analysis. To the best of our knowledge, there are currently no established methods to account for measurement error in GAMLSS. As a result, the uncertainty reported in our study may underestimate the actual uncertainty. (iii) Studies such as Pullum et al.[5] and Seiler et al.[50] highlight that the included response variables may be skewed and have heavier distribution tails, so relating the responses based on a multivariate Gaussian distribution can be seen as a starting point. In practice, more flexible multivariate distributions would potentially improve the estimates by reducing the introduced bias, especially

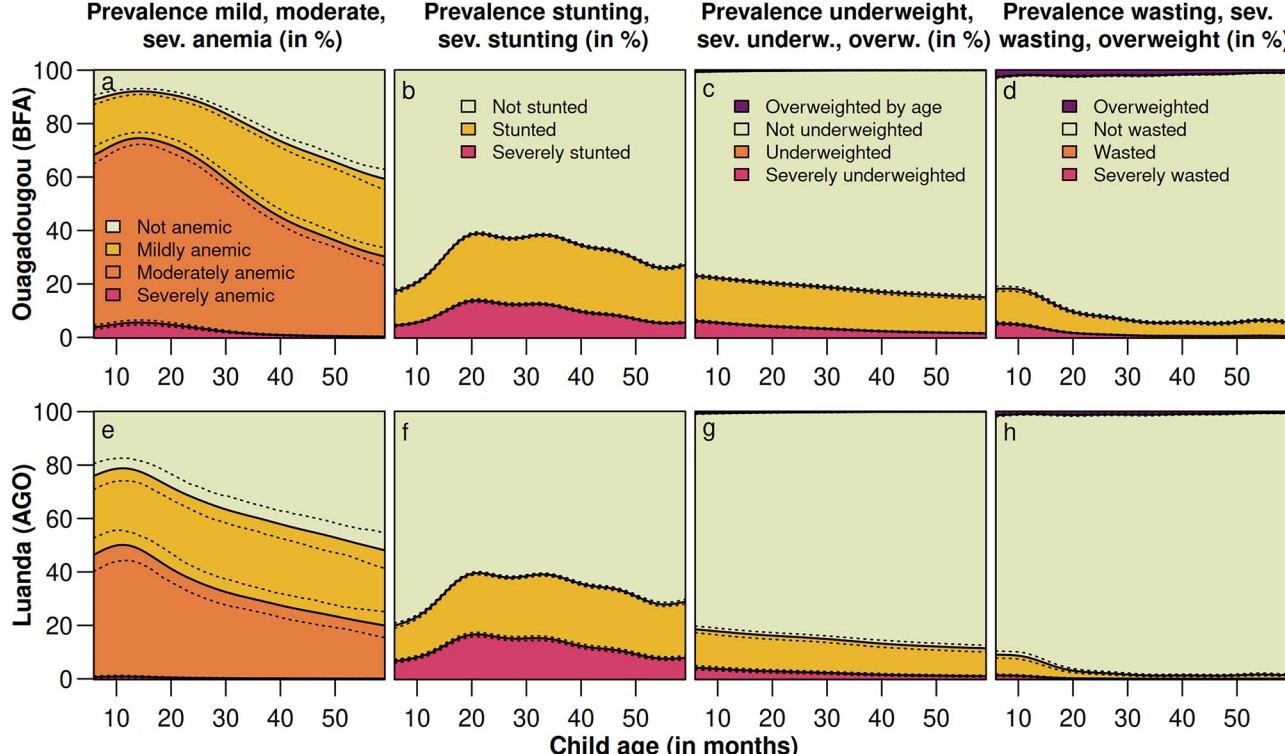

**Fig. 4 | Estimated marginal prevalence of anemia, stunting, underweight, wasting, and overweight among children between 6 and 59 months by age for selected locations. a–d** prevalence estimates based on the Hb level, HAZ, WAZ, and WHZ in Ouagadougou; and **e–h** prevalence estimates based on the Hb level, HAZ, WAZ, and WHZ in Luanda. To estimate the marginal prevalence, values for all other numeric covariates are fixed at the median observed within a buffer of 20 km around the location, and values for categorical covariates are fixed at the mode observed within the buffer around the given location, respectively. The country abbreviations correspond to the country-specific ISO3 country codes. Note that these estimates are based on the final model, which was fitted using the training data ($n = 164,300$) and validated using the test data ($n = 41,074$). For more details, see *Supplementary Fig.* 1.

for extreme quantiles. However, to date, there is no out-of-the-box solution to incorporate more complex multivariate distributions. (iv) It should be noted that if the focus of the analysis is on a single nutrition-related indicator, it is more appropriate to model this indicator independently. This would allow for a larger sample size and greater flexibility in the choice of response distribution, potentially leading to more precise and robust estimates. However, the strength of our approach lies in the ability to examine the co-occurrence of multiple indicators.

By acknowledging these limitations, we aim to provide a balanced interpretation and to highlight the importance of ongoing efforts to improve data quality and advance methodological approaches in studies of this type. Such efforts are essential to provide policymakers with the insights they need to effectively target vulnerable populations and allocate public health resources to areas of greatest need.

The presented country-level prevalence estimates for 2005 to 2020 align closely with other data sources, such as the UNICEF/WHO Joint Malnutrition Estimates[66] and WHO Global Health Observatory[7]. However, our estimates reveal significant subnational variability in prevalence and correlation. This underscores the importance of high-resolution monitoring, as applied in prior univariate studies[46,48,50]. The modest correlation between Hb and anthropometric indicators (i.e., $\rho \approx 0.00$–$0.18$) aligns with previous studies from different geographic target areas[69], underlining the weak overlap and reinforcing calls to complement growth monitoring with Hb screening. In contrast, the very high correlations between WAZ and WHZ (i.e., $\rho \approx 0.63$–$0.77$) and HAZ and WAZ (i.e., $\rho \approx 0.60$–$0.74$) support earlier findings[22–24], suggesting that interventions for wasting are likely to affect the prevalence of underweight and stunting concurrently. This study highlights and supports previous findings[26–29,71,72] that anemia and wasting peak between nine and 18 months, while stunting increases with age,

peaking between the second and third years of life and persisting at high levels until age five. This underscores the need for micronutrient and infection control packages during the first 1000 days combined with catch-up growth programs during the preschool years.

Identifying spatial co-occurrence patterns, along with the common risk factors associated with its key determinants, offers valuable insights for policymakers. The results aim to guide the allocation of scarce resources and targeted interventions that are essential to improving the quality of life for young children in Africa, especially in regions facing a high double or triple burden of malnutrition. These efforts not only address immediate health challenges but also play a critical role in enhancing long-term health outcomes, educational attainment, and economic productivity. The findings point to the following critical policy implications: (i) *Sharper geographic targeting of scarce resources:* Identifying hotspots helps policymakers, NGOs, and Ministries of Health concentrate e.g., fortified blended foods, micronutrient powders, and climate-resilient social protection cash transfers in these areas first rather than spreading programs nationwide. (ii) *Bundling interventions:* The strong positive correlations among HAZ, WAZ, and WHZ, as well as their overlap with anemia, combined with the high prevalence of these conditions, demonstrate that children often experience multiple deficits. Rather than running separate anemia, wasting, stunting, and overweight programs, governments can implement bundled interventions (e.g., anti-malarial bed net campaigns, fortification, school-based nutrition programs, improving access to healthier foods, growth monitoring, and WaSH programs) in co-burden areas, reducing delivery costs and beneficiary fatigue. (iii) *Addressing the age gradient:* The age gradient shows that anemia and wasting peak before 24 months of age, while stunting is lowest in young children below one year of age. Therefore, counseling on infant and young child feeding, fortified blended foods,

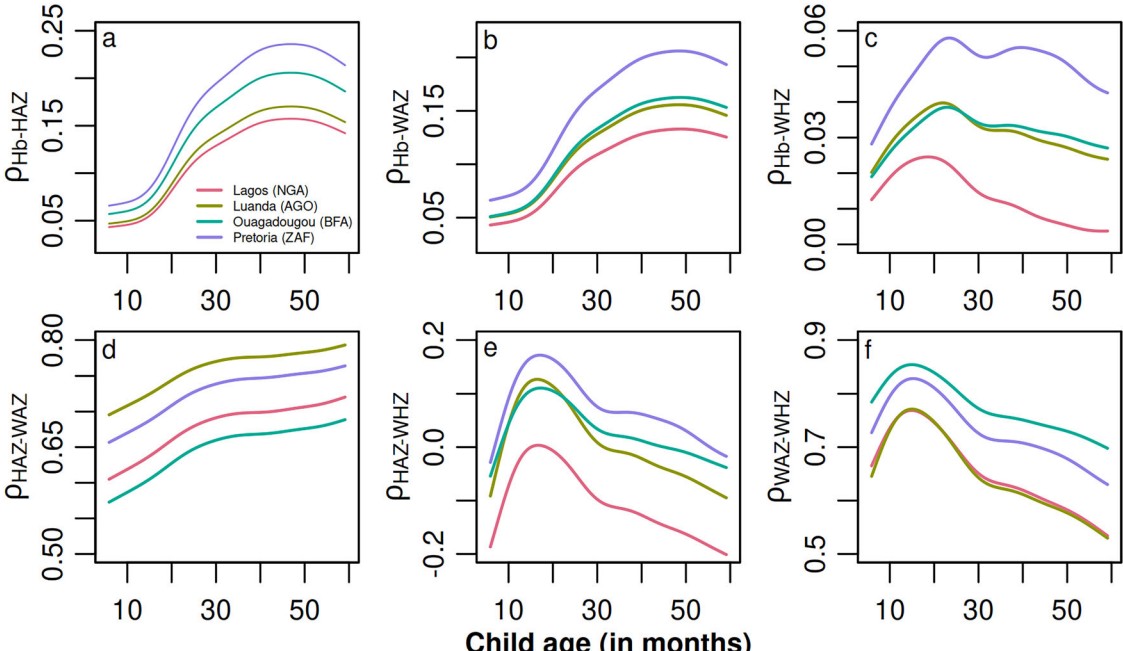

**Fig. 5 | Estimated pairwise correlation between four key nutritional indicators by age at four selected locations. a** Hb level and HAZ, 2020; **b** Hb level and WAZ, 2020; **c** Hb level and WHZ, 2020; **d** HAZ and WAZ, 2020; **e** HAZ and WHZ, 2020; and **f** WAZ and WHZ, 2020. To estimate the pairwise correlation, values for all other numeric covariates are fixed at the median observed within a buffer of 20 km around the location, and values for categorical covariates are fixed at the mode observed within the buffer around the given location, respectively. The country abbreviations correspond to the country-specific ISO3 country codes. Note that these estimates are based on the final model, which was fitted using the training data ($n = 164,300$) and validated using the test data ($n = 41,074$). For more details, see *Supplementary Fig.*1.

micronutrient powders, and deworming programs should focus on the first 1000 days. (iv) *Continuous updating:* Because the model produces prevalence estimates, governments and national statistical offices can incorporate them into their routine DHS reanalyses to update maps. This allows for more frequent tracking of progress toward *SDG* 2 and aligns domestic targets with the UN's *leave no one behind* principles. (v) *Early warning-system integration:* Spatial covariates (e.g., temperature and precipitation anomalies, or malaria incidence) have emerged as significant predictors. This information can be used to predict where seasonal surges are expected and can be integrated into an early-warning system.

## Data availability

All data used in the analysis of the main text and Supplementary Information are available from the cited sources (see *Supplementary Table*1). *Supplementary Table*1 also provides URLs and references to the data sources in the last column. Note that all of the data sources cited in *Supplementary Table*1 are freely available, however, although freely available after registration and approval, we do not have the permission from ICF International Inc. to redistribute DHS data. The underlying data to reproduce Fig. 1 to Fig. 5 of the manuscript is provided in *Supplementary Data*1 (i.e., *Supplementary_Data_1.zip*) to *Supplementary Data*3 (i.e., *Supplementary_Data_3.zip*). The figures can be plotted using the R-script figures.R available in a Zenodo repository: https://doi.org/10.5281/zenodo.18087416[74].

## Code availability

Software The results of this paper have been accomplished using custom software, tailored to be used at the high performance (HPC) infrastructure *LEO* of the University of Innsbruck. For that purpose the statistical software R[73] using the following R packages has been used: bamlss[65,75,76], backports[77], broom[78], coda[79], codetools[80], colorspace[81,82], deldir[83], dismo[84], gamlss.dist[85], maps[86], mgcv[87,88], mvnchol[51], mvtnorm[89], nlme[90,91], pillar[92], raster[93], rgeos[94], rgdal[95], rnaturalearth[96], rnaturalearthdata[97], rnaturalearthhires[98], scales[99],

scoringRules[100], sf[101,102], smoothr[103], sp[104,105]. The custom R-code used for statistical analysis is freely available in a Zenodo repository: https://doi.org/10.5281/zenodo.18087416[74]. This repository also includes a *README* file containing further information on how to use the code. Computational details The custom computer code is freely available in a Zenodo repository (https://doi.org/10.5281/zenodo.18087416[74]). Please note that the code is custom tailored to the HPC infrastructure *LEO* of the University Innsbruck and adaptions to other systems may be required.

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

## Acknowledgements

The computational results presented have been achieved (in part) using the HPC infrastructure LEO of the University of Innsbruck. We thank ICF International, Inc. and USAID for conducting the DHS, and providing public access to the data. We are grateful to the three anonymous reviewers for their critical yet constructive feedback that led to substantial improvements in the paper. This research was funded in whole/in part by the Austrian Science Fund (FWF) grant https://doi.org/10.55776/P33941. For open access purposes, the authors have applied a CC BY public copyright license to any author accepted manuscript version arising from this submission. J.S. gratefully acknowledges the funding of the Vice-Rectorate for Research of the University of Innsbruck (# 419533). Software: The computational results presented have been derived using the statistical software R[73]. For details, see the code availability statement. Data and materials availability: All data used to conduct the analysis of the main text and the *Supplementary Information* is available under the cited references.

## Author contributions

Conceptualization: J.S., K.H.; Methodology: J.S., B.M., M.W., N.U.; Statistical analysis: J.S., B.M., N.U., R.S.; Visualization: J.S., B.M., R.S.; Funding acquisition: J.S., K.H., N.U.; Project administration J.S., K.H.; Supervision: K.H., I.G., N.U.; writing–original draft: All authors. All authors provided essential input, throughout all stages of the drafting of the manuscript.

## Competing interests

The authors declare no competing interests.
