## [Transparent Peer Review file · Communications Medicine]

Co-occurrence patterns of malnutrition indicators among children in sub-Saharan Africa

Corresponding Author: Dr Johannes Seiler

Version 0:

Reviewer comments:

Reviewer #1

(Remarks to the Author)

Overview of the article:

This manuscript used a multivariate Gaussian model to estimate co-occurrence patterns, shared risk factors, and spatiotemporal patterns of four nutrition indicators (haemoglobin, HAZ, WAZ, and WHZ) among children 6 to 59 months. Data from Demographic and Health Surveys were used for 30 countries between 2003 and 2020. Data on climate, demographics and environmental factors were also included.

The rationale for this work is explained to be that malnutrition is a major risk factor for child mortality, especially in South Asia and sub-Saharan Africa. Also, other than monitoring progress to SDG 2 (and 3?), this work is considered to add to knowledge to inform public health strategies, nutritional interventions and overall health.

The three objectives and their findings are as follows:

1. Analysing the co-occurrence patterns, levels, and trends of anaemia, acute malnutrition, and chronic malnutrition.
 - a. Correlations ranged between -0.10 (WHZ & HAZ) and 0.68 (WAZ & WHZ).
 - b. Variations between countries outweigh temporal variation.
 - c. Correlation between Hb and anthro: Low (0.05 to 0.24)
 - d. Correlation between anthro indicators: high (0.87).
 - e. Negative correlation between WHZ and HAZ?
 - f. Correlation between HB and Anthro is highest in Uganda, Ghana, Cote d'Ivoire and Mozambique.
2. Identify shared risk factors associated with these conditions.
 - a. Main risk factors: age of the child and geographic location.
 - i. Age: Anaemia and acute and chronic malnutrition are not evenly distributed across the age range.
 - ii. Location: Anaemia, and acute and chronic malnutrition often coincide in location.
 - b. Other covariates: child's sex, household wealth index, socioeconomic and environmental characteristics (surface temperature, immunisation status, GDP, malaria incidence). Other aspects highlighted by this:
 - i. Household characteristics.
 - ii. Influence of environmental factors.
 - c. Shared risk factors:
 - i. Prevalence of anaemia, underweight, and wasting is highest at young ages and decreases with age.
 - ii. Chronic malnutrition is lowest in young children and then increases with age.
3. Investigate spatiotemporal dynamics and patterns across LMICs in SSA ...
 - a. (Provided) – but little said about this?

Discussion: The limitations are mentioned, as well as the overall (broad) contribution of this work.

Feedback to authors:

Congratulations on a well-written and interesting article. The research gap for this work is clearly indicated. However, I will recommend the following adjustments to add to the strength of this work and to ensure its worth to a practitioner.

1. You do provide a good literature background and rationale for the work. In it, you also refer to the overweight rates. This

lets the reader think that this will also be an aspect which you will address in this article. I will recommend, as this is not the case, that you do not focus on overweight and obesity in the background section, but keep the focus on undernutrition.

2. You do refer to SDG3 in the literature background. But, other than the SDG2, it is not indicated what the SDG aims for and hence what the relevance of SDG3 is in this work.
3. Thank you for clearly providing the objectives of this work. However, I did feel that this could be deliberated a bit more in the text.
 - a. Objective 1: the co-occurrence patterns are provided, but very little is said about the levels and trends. Although it is provided in the figures, I thought a bit more could be added to the text.
 - b. Objective 2: the shared risk factors are mentioned, but this is very broad. An example is the household characteristics— which characteristic should be noted? What about these characteristics? What influence do they have? In line 148 – 149 it is indicated that ‘these findings’ will be examined in more detail in the following section. I though missed it / or it was not clear enough.
 - c. Objective 3: provided, but little said in the text?
4. Figure 4: Not referred to in the text.
5. I am missing an in-depth discussion placing the findings in context with other studies/ knowledge. Although I do find the findings very interesting, I am missing an interpretation of the findings and the ‘so what’.

Reviewer #2

(Remarks to the Author)

this paper studies the joint distribution of anemia, acute, and chronic malnutrition in sub-Saharan Africa using a multivariate Gaussian regression model.

The major claim of the paper is that co-occurrences studies as the present one are of high importance for policy makers and public health policies towards the nutrition-related Sustainable Development Goal 2 (SGD2). However, the authors use a regression model which cannot provide cause-effect relationships and hence the results can be of descriptive nature only. They have to be taken with caution.

It is also not clear on how the results would directly influence or change the approach to reach SGD2. Hence, the direct practical implications and benefits remain unclear.

I believe that spatial confounding may be an issue in the data. Did the authors investigate this?

To SGD2 being effective I'd expect to provide confidence/p-values or some measures of uncertainty. This was not considered by the authors.

can the employed two-step approach (first selecting the covariates and then fitting the final model) lead to consistent estimators?

Reviewer #3

(Remarks to the Author)

This study analyzes co-occurrence patterns of key nutrition indicators in sub-Saharan Africa, revealing significant variability and strong correlations, emphasizing the need for localized, multi-sectoral strategies to improve child health outcomes. The paper is generally very well written. The authors appear to have used robust and fit for purpose data as well as analytical approaches. My major concern is the policy utility and that the paper in its current format does not provide output that will be readily useful for local policy makers to make spatially targeted decisions.

While some notable results are presented, I think the paper could be repurposed to provide output that is more useful for local policy/decision makers.

Methods - lacking details related to model validation. This should be included.

Methods - lacking description of how the continuous spatial surfaces were created (prediction at unsampled locations), as in Fig 2/3. Apologies if I missed this.

Methods - did the authors consider using a spatial-temporal clustering approach (e.g. Kulldorff space time statistic) to identify significant high and low aggregations among the survey data points? Characteristics within these clusters could then also be compared to those outside of the hot or cold spots.

Results, Figure 4 - please consider overlaying the raw data points on these curves to show how well they fit.

Results, Figure 5 - not sure of the utility of this, especially from policy/decision maker perspective. The patterns are quite similar and age is proxy for other underlying proximal risk factors.

Results - it would also be useful to incorporate metrics related to country/subnational progress towards SDG 2.2 (2030). As you incorporate year in your model formulation, this could be summarised based on raw datapoints by year compared with model prediction by year.

Discussion - more detailed and specific recommendations/implications of the research should be incorporated. This should also include key recommendations by country/sub-national within country.

Version 1:

Reviewer comments:

Reviewer #1

(Remarks to the Author)

I appreciate the thorough revisions. I am satisfied with the corrections made. The authors have addressed my comments appropriately, and the revised manuscript meets my expectations. The paper is will be of interest to the field.

Reviewer #2

(Remarks to the Author)

Reviewer #3

(Remarks to the Author)

I would like to commend the authors on their very thorough, detailed, and considered responses to my comments - excellent work.

Discretionary: Regarding my previous comment on Figure 4 (about overlaying raw data points to show fit), I fully understand that multivariable adjustment can alter the relationship between the modelled age-specific fractions and the underlying raw survey data. It may nonetheless be helpful to overlay the observed survey fractions by age and location as a broad reference - these could be dots for underlying survey fraction relating to each shaded portion. This would provide readers with a visual check that the modelled fractions are, at least approximately, consistent with the underlying data.

Response to the *Comments Raised by the Reviewers* for the article:
*Co-occurrence patterns of malnutrition indicators among children in
sub-Saharan Africa (COMMSMED-24-1333-T)*

submitted to

Communications Medicine

August 14, 2025

Comments Reviewer #1 (expertise: nutrition and dietetics)

Overview of the article

This manuscript used a multivariate Gaussian model to estimate co-occurrence patterns, shared risk factors, and spatiotemporal patterns of four nutrition indicators (haemoglobin, HAZ, WAZ, and WHZ) among children 6 to 59 months. Data from Demographic and Health Surveys were used for 30 countries between 2003 and 2020. Data on climate, demographics and environmental factors were also included. The rationale for this work is explained to be that malnutrition is a major risk factor for child mortality, especially in South Asia and sub-Saharan Africa. Also, other than monitoring progress to SDG 2 (and 3?), this work is considered to add to knowledge to inform public health strategies, nutritional interventions and overall health.

The three objectives and their findings are as follows:

1. Analysing the co-occurrence patterns, levels, and trends of anaemia, acute malnutrition, and chronic malnutrition.
 - (a) Correlations ranged between -0.10 (WHZ & HAZ) and 0.68 (WAZ & WHZ).
 - (b) Variations between countries outweigh temporal variation.
 - (c) Correlation between Hb and anthro: Low (0.05 to 0.24)
 - (d) Correlation between anthro indicators: high (0.87).
 - (e) Negative correlation between WHZ and HAZ?
 - (f) Correlation between HB and Anthro is highest in Uganda, Ghana, Cote d'Ivoire and Mozambique.
2. Identify shared risk factors associated with these conditions.
 - (a) Main risk factors: age of the child and geographic location.

- i. Age: Anaemia and acute and chronic malnutrition are not evenly distributed across the age range.
 - ii. Location: Anaemia, and acute and chronic malnutrition often coincide in location.
 - (b) Other covariates: child’s sex, household wealth index, socioeconomic and environmental characteristics (surface temperature, immunisation status, GDP, malaria incidence). Other aspects highlighted by this:
 - i. Household characteristics.
 - ii. Influence of environmental factors.
 - (c) Shared risk factors:
 - i. Prevalence of anaemia, underweight, and wasting is highest at young ages and decreases with age.
 - ii. Chronic malnutrition is lowest in young children and then increases with age.
3. Investigate spatiotemporal dynamics and patterns across LMICs in SSA ...
- (a) (Provided) – but little said about this?

Response We thank Reviewer #1 for the valuable input, critical feedback, and accurate summary of our article. In the revised version of the manuscript, we focused on spatio-temporal dynamics of prevalence, as well as pairwise correlations. We also discuss policy implications and the other aspects highlighted by the three reviewers.

Feedback to authors:

Congratulations on a well-written and interesting article. The research gap for this work is clearly indicated. However, I will recommend the following adjustments to add to the strength of this work and to ensure its worth to a practitioner.

Response We appreciate the critical feedback, which strongly helped us to improve the manuscript. Please find below our detailed, point-by-point response to the comments raised by Reviewer #1. We have addressed each comment with great care and are confident that we have addressed all of them in a satisfactory manner.

1. You do provide a good literature background and rationale for the work. In it, you also refer to the overweight rates. This lets the reader think that this will also be an aspect which you will address in this article. I will recommend, as this is not the case, that you do not focus on overweight and obesity in the background section, but keep the focus on undernutrition.

Response We acknowledge the concern about the focus of the background section. Using generalized additive models for location, scale, and shape (GAMLSS; [1]), prevalence estimates of overweight based on weight-for-height (WHZ) or weight-for-age (WAZ) z-scores can easily be incorporated straightforwardly, as the full distribution of the underlying continuous covariates is modeled. In the context of public health in the region, we consider overweight to be a relevant and increasingly important issue. Furthermore, overweight constitutes to be a form of malnutrition. Due to changes in food intake and a shift toward calorie-dense, less nutritious foods, the issue of overweight needs to be addressed from a public health perspective as it contributes to the double and triple burden of malnutrition. Consequently, we now discuss the results of overweight

(based on the WHZ) and its policy implications alongside the more *classical* nutritional indicators. Furthermore, as shown in *Figure 2*, as well as *Supplementary Figures 6 to 11*, we present high-resolution estimates of overweight alongside those for anemia, stunting, underweight, and wasting.

2. You do refer to SDG3 in the literature background. But, other than the SDG2, it is not indicated what the SDG aims for and hence what the relevance of SDG3 is in this work.

Response Thank you for pointing out this omission. In the revised literature background, we discuss the relevance of this work with respect to *SDG 2* (Zero Hunger) and *SDG 3* (Good Health and Well-being). Addressing malnutrition is crucial to reducing preventable child deaths. Previous studies estimate that malnutrition is responsible for 35% to 45% of post-neonatal and under-five deaths, highlighting the implications of *SDG 2* on *SDG 3* [2–5]. By analyzing the co-occurrence patterns of malnutrition indicators, we also contribute to a better understanding of the underlying causes of child mortality, which can improve overall health outcomes.

This link is now reflected in the literature background, where we added the following paragraph:

L32 – L51: The observed patterns of malnutrition are closely linked to child mortality, as malnutrition is considered a major risk factor for child mortality [2, 3, 6–8]. It is estimated that approximately 35% to 45% of all child deaths globally among children under five years – nearly half of all child deaths – can be attributed to some form of malnutrition [2–5]. Among other factors, malnutrition weakens the immune system and impairs growth and development, making anemia and undernutrition major contributors to infant and under-five mortality. For instance, anemia, particularly due to iron and vitamin deficiency, further increases children’s vulnerability and increases the risk of infections and complications, which can also lead to a higher risk of early child death. Despite substantial global progress, infant and under-five mortality rates remain high in some regions, particularly in SSA, where the estimated infant (under-five) mortality rate was 4.4% (6.8%) in 2023. These rates are substantially higher than the reported infant (under-five) mortality rate in South Asia, the second most affected region, where the rate was 3.0% (4.5%), and about nine times higher than in North America in the same year. In absolute terms, about two million children under the age of one died in SSA, accounting for about 50% of all global child deaths in 2023 [8]. Understanding and effectively addressing the nutritional aspects of early childhood development is therefore not only a matter of meeting *SDG 2*, but also critical to *SDG 3*, which aims to ensure healthy lives and promote well-being for all at all ages, including specific targets to reduce neonatal and under-five mortality in all countries by 2030.

3. Thank you for clearly providing the objectives of this work. However, I did feel that this could be deliberated a bit more in the text.

Response We appreciate the reviewer’s positive feedback on the clarity of our objectives and the suggestion to elaborate further. In the revised manuscript, we updated the introduction to more clearly articulate the rationale behind our objectives and their relevance within the broader research context. Additionally, we expanded the Results and Discussion sections to explicitly connect our findings to practical implications, particularly in the context of monitoring and addressing *SDG 2* (Zero Hunger) and *SDG 3* (Good Health and Well-Being). These revisions highlight how our work contributes to the academic understanding of the co-occurrence patterns of anemia and other indicators of malnutrition in SSA and its implications for policies and programs targeting these critical global development goals.

- (a) Objective 1: the co-occurrence patterns are provided, but very little is said about the levels and trends. Although it is provided in the figures, I thought a bit more could be added to the text.

Response The revised manuscript now includes the paragraph *Estimated prevalence and trends of anemia, stunting, underweight, wasting, and overweight* (L266 – L326), which provides details on the levels and trends of all five indicators within the included countries in SSA.

Additionally, we would like to clarify that individual prevalence estimates are generally more robust when derived from univariate models because these models allow for greater flexibility in specifying distributional assumptions and spatial structures. In contrast, the incorporated multivariate model is optimized for assessing co-occurrence patterns. It does not include all spatial terms for each outcome, which may modestly influence the marginal prevalence estimates. Therefore, we restrict our discussion of prevalence results to the country level and provide them primarily for contextualizing the co-occurrence analysis.

- (b) Objective 2: the shared risk factors are mentioned, but this is very broad. And example is the household characteristics – which characteristic should be noted? What about these characteristics? What influence do they have? In line 148 – 149 it is indicated that ‘these findings’ will be examined in more detail in the following section. I though missed it / or it was not clear enough.

Response We apologize for this inconvenience. Upon reviewing the initial submission, we found that parts of this section were accidentally deleted during the typesetting process. In the revised version of the manuscript (L358 – 395), we paid close attention to the section on shared risk factors and ensured that no similar typesetting errors occurred.

- (c) Objective 3: provided, but little said in the text?

Response We now provide additional details on the spatio-temporal dynamics of the individual indicators and their co-occurrence patterns in the paragraphs labeled *Estimated prevalence and trends of anemia, stunting, underweight, wasting, and overweight* (L266 – L326) and *Patterns of co-occurrence* (L327 – L356). We ask the reviewer to refer directly to these paragraphs. Additionally, please refer to *Figure 2* and *Figure 3* in the main manuscript, as well as *Supplementary Figures 6 to 14*, which present temporal trends in anemia, stunting, underweight, wasting, and overweight, along with changes in pairwise correlations over time.

4. Figure 4: Not referred to in the text.

Response Thank you for pointing this out. We carefully checked the manuscript to ensure that all figures and tables were referred and described correctly.

5. I am missing an in-depth discussion placing the findings in context with other studies/ knowledge. Although I do find the findings very interesting, I am missing an interpretation of the findings and the ‘so what’.

Response We thank Reviewer #1, Reviewer #2, and Reviewer #3 – who highlighted similar aspects – for raising this important point. We now provide a literature synthesis in the *Discussion* (L433 – L448), focusing on the following aspects: (i) put our results into perspective, and compare our results (prevalence and co-occurrence metrics) to existing results; (ii) interpret the age-specific

and spatial correlation patterns in light of existing studies; and (iii) better articulate the novel scientific contributions to the empirical strand of literature.

The revised *Discussion* now explains how the results, especially the high-resolution risk maps, can provide actionable guidance for ministries and implementing partners. Although the work is not causal in nature, the results directly influence the approach to achieving *SDG 2*. Specifically, this is achieved through the following points mentioned in the discussion (L449 – L477): (i) *Sharper geographic targeting of scarce resources*; (ii) *Bundling interventions*; (iii) *Addressing the age gradient*; (iv) *Continuous updating*; (v) *Early warning-system integration*.

We would like to thank Reviewer #1 for the overall positive review. Especially the detailed, constructive and yet challenging feedback is very much appreciated. It helped us a lot to improve the current quality of the article. We hope that we have been able to address all of Reviewer #1's points adequately.

Comments Reviewer #2 (expertise: statistics and computational methods)

This paper studies the joint distribution of anemia, acute, and chronic malnutrition in sub-Saharan Africa using a multivariate Gaussian regression model.

The major claim of the paper is that co-occurrences studies as the present one are of high importance for policy makers and public health policies towards the nutrition-related Sustainable Development Goal 2 (SGD2). However, the authors use a regression model which cannot provide cause-effect relationships and hence the results can be of descriptive nature only. They have to be taken with caution.

Response We thank Reviewer #2 for the helpful comment regarding the interpretation and implications of our results. We acknowledge that this study is not designed to establish causal relationships between covariates and the co-occurrence of different indicators of malnutrition. Accordingly, our results should be interpreted as descriptive in nature, offering insights into spatial patterns and associations rather than causal effects.

This approach is consistent with a well-established strand of the literature that prioritizes precise spatio-temporal estimation of underlying health indicators rather than causal identification (see, e.g., [9–13]). In such studies, the focus is on generating reliable, high-resolution maps of health outcomes to inform programmatic action, rather than on disentangling the causal mechanisms behind observed associations.

We believe that identifying geographic hot spots and cold spots where multiple nutrition-related indicators are simultaneously poor or favorable remains highly valuable for policymakers. Such information can support the effective allocation of limited resources and guide targeted interventions – both of which are essential for advancing progress toward *SDG 2* and *SDG 3*.

To reinforce the descriptive nature of our findings, we have added a clarifying statement at the beginning of the *Discussion* section and expanded our comments on the limitations of the study (L397 – L432).

It is also not clear on how the results would directly influence or change the approach to reach SGD2. Hence, it the direct practical implications and benefits remain unclear.

Response We thank Reviewer #2, Reviewer #1, and Reviewer #3 – who highlighted similar aspects – for raising this important point. We now provide a literature synthesis in the *Discussion* (L433 – L448), focusing on the following aspects: (i) put our results into perspective, and compare our results (prevalence and co-occurrence metrics) to existing results; (ii) interpret the age-specific and spatial correlation patterns in light of existing studies; and (iii) better articulate the novel scientific contributions to the empirical strand of literature.

The revised *Discussion* now explains how the results, especially the high-resolution risk maps, can provide actionable guidance for ministries and implementing partners. Although the work is not causal in nature, the results directly influence the approach to achieving *SDG 2*. Specifically, this is achieved through the following points mentioned in the discussion (L449 – L477): (i) *Sharper geographic targeting of scarce resources*; (ii) *Bundling interventions*; (iii) *Addressing the age gradient*; (iv) *Continuous updating*; (v) *Early warning-system integration*.

I believe that spatial confounding may be an issue in the data. Did the authors investigate this?

Response We thank Reviewer #2 for raising the important issue of spatial confounding. We have addressed this concern explicitly in our modeling approach. Specifically, we included a smooth spatial effect to capture unmeasured spatial variation, which could otherwise confound the relationships

between the covariates and response variables (i.e., co-occurrence patterns of different indicators of malnutrition).

To address spatial variation over time as well as for different ages, the spatial component interacts with the interview year and the child’s age. These interactions are important as spatial patterns can change over time, and children of different ages may be affected by different environmental or socioeconomic factors, which can impact children’s vulnerability to malnutrition in spatially structured ways.

Additionally, we have included a comprehensive set of spatially varying covariates derived from remote sensing and other geospatial data sets. These covariates include 2 m surface temperature anomalies and malaria incidence, among others. A complete list of the included covariates and their model specifications can be found in *Supplementary Table 3*. These covariates are designed to account for known environmental and contextual factors that could influence the outcome.

Our approach aligns with the methodologies commonly used in high-resolution health mapping studies (e.g., [9–13]). These studies consider the use of flexible spatial random effects and the inclusion of spatially varying covariates to be the gold standard for mitigating spatial confounding. See *Equation (4)* in *Supplementary Note 2* for further clarification on how spatial and covariate terms are incorporated in the model.

To SGD2 being effective I’d expect to provide confidence/p-values or some measures of uncertainty. This was not considered by the authors.

Response The final model is now estimated using Bayesian simulation techniques, which allows for an assessment of the uncertainty in the estimated effects. Where applicable, the corresponding uncertainty estimates have been added (see e.g., *Figure 4*, *Supplementary Figure 6 to 8*) and are now also discussed in the manuscript. Our updated analytical approach consists of the following five steps (see *Supplementary Note 2* for details): (i) *Creating the training and test data set*; (ii) *Estimating a baseline reference model*; (iii) *Identifying informative covariates in each predictor η_k* ; (iv) *Bayesian estimation of the final model omitting uninformative covariates*; (v) *Model validation*.

Can the employed two-step approach (first selecting the covariates and then fitting the final model) lead to consistent estimators?

Response The applied two-step approach first selects informative covariates and then uses Bayesian simulation techniques to estimate the final model. We recognize the importance of ensuring this approach results in consistent estimators, and we appreciate the opportunity to address this concern. In general, post-selection estimation can lead to biased or inconsistent estimators if the uncertainty from the covariate selection step is ignored. Asymptotic consistency of the final estimators requires that: (i) the selection procedure consistently identifies the true set of covariates as the sample size increases, (ii) the final model is correctly specified, and (iii) priors are well-behaved. If these conditions hold, then the two-step approach yields consistent estimators.

In our case, the boosting-like selection method used in the first step is not guaranteed to be selection-consistent in a strict asymptotic sense, particularly when the number of covariates is large relative to the sample size. However, given the large sample size of approximately 160,000 observations and the moderate number of candidate covariates in our application, the risk of omitting important predictors is expected to be extremely small in practice.

For the second step, Bayesian simulation techniques produce consistent estimators under correct model specification and appropriate priors (see *Supplementary Note 2.2* for details). The chosen priors have been shown to perform well in terms of efficiency and robustness in numerous applications [13–18].

While finite-sample bias due to imperfect variable selection cannot be entirely ruled out, we mitigate

this risk through cross-validation to reduce overfitting and selection bias. Consequently, although the approach may not provide exact post-selection inference, we expect the point estimates to be asymptotically consistent and practically reliable for the current study.

We thank Reviewer #2 for the positive review and for the detailed, constructive yet sometimes challenging feedback. This feedback helped us improve the quality of the current manuscript. We hope we have adequately addressed all the points raised by Reviewer #2 and the other two anonymous reviewers.

Comments Reviewer #3 (expertise: epidemiology and biostatistics)

This study analyzes co-occurrence patterns of key nutrition indicators in sub-Saharan Africa, revealing significant variability and strong correlations, emphasizing the need for localized, multi-sectoral strategies to improve child health outcomes. The paper is generally very well written. The authors appear to have used robust and fit for purpose data as well as analytical approaches. My major concern is the policy utility and that the paper in its current format does not provide output that will be readily useful for local policy makers to make spatially targeted decisions.

While some notable results are presented, I think the paper could be repurposed to provide output that is more useful for local policy/decision makers.

Response We thank Reviewer #3 for the thoughtful summary of our manuscript and the constructive feedback, both of which have been invaluable in improving the clarity and utility of our paper. In response to concerns about the utility to policymakers, we have included detailed high-resolution prevalence estimates for anemia, stunting, underweight, wasted, and overweight (based on WHZ). These estimates are useful for identifying hot spots and cold spots, which can inform more targeted interventions and help allocate scarce resources efficiently. By providing these spatially explicit estimates, we believe that our analysis now offers more actionable insights that local policymakers can use to tailor interventions to specific areas. We hope this strengthens the paper’s practical relevance. Please see our detailed, point-by-point responses to the comments below.

Methods – lacking details related to model validation. This should be included.

Response We have included more detailed information about the model validation process in the revised manuscript. Please see Section *Methods* of the manuscript and *Supplementary Note 2*, particularly *Supplementary Note 2.2*. Specifically, we split the data into training and test data sets while accounting for its spatial structure. This ensures that the model is validated in a way that reflects the geographic structure of the data. Additionally, we performed extensive cross-validation to assess the model’s generalizability and identify potential sparseness issues related to the data. These details have been incorporated into the *Methods* section and are described in detail in *Supplementary Note 2*.

Methods – lacking description of how the continuous spatial surfaces were created (prediction at unsampled locations), as in Fig 2/3. Apologies if I missed this.

Response We completely revised the Methods section to make the underlying methodological aspects as transparent as possible. To clarify, predictions at unsampled locations, i.e., the continuous spatial surfaces, are based on the final model estimates. The estimation routine incorporates five steps: (i) *Creating the training and test data set*; (ii) *Estimating a baseline reference model*; (iii) *Identifying informative covariates in each predictor η_k* ; (iv) *Bayesian estimation of the final model omitting uninformative covariates*; (v) *Model validation*. In *Step 1*, carefully creating the training and test data sets ensures that the test data are spatially representative. This step is critical because poor spatial representation can lead to estimation errors and unreliable uncertainty estimates. Predictions for unsampled locations are based on the final multivariate model from *Step 4* and the corresponding covariate values for those locations. In *Step 5*, the model is validated through rigorous procedures, including cross-validation, to check for data sparsity issues and ensure reliable estimates. See *Supplementary Note 2.2* for details. These diagnostic checks give us confidence that our predictions at unsampled locations are valid.

Methods – did the authors considering using a spatial-temporal clustering approach (e.g. Kulldorff space time statistic) to identify significant high and low aggregations among the survey data points? Characteristics within these clusters could then also be compared to those outside of the hot or cold spots.

Response Given the continuous nature of space (longitude and latitude) and outcomes (hemoglobin levels and anthropometric z-scores), as well as the large sample size, we believe that generalized additive models for location, scale, and shape (GAMLSS) [1] are better suited for our analysis than spatial scan statistics.

While the original Kulldorff space–time statistic is designed for discrete count or binary outcomes (e.g., Poisson or Bernoulli data), extensions to continuous outcomes do exist. However, such methods still focus on detecting discrete clusters and require exhaustive searches over spatial and temporal windows. This involves repeatedly evaluating likelihoods for a large number of candidate regions, which becomes computationally prohibitive for our dataset of approximately 160,000 observations. For example, preliminary attempts using the R package **HDSpatialScan** [19, 20] resulted in memory demands exceeding available resources and runtimes of several hours without completion.

In contrast, GAMLSS models can flexibly incorporate smooth spatial effects (functions of longitude and latitude) and interactions with other predictors, allowing us to capture gradual spatial variation and space–time interactions directly. This is crucial for our objectives, as it enables estimation of continuous spatial surfaces rather than detection of discrete clusters. Moreover, the computational demands of GAMLSS are considerably lower for datasets of this size, making the approach more practical.

In summary, while spatial scan statistics are valuable for discrete cluster detection in count or binary data, their computational burden and discrete-cluster focus make them less suitable for our large, continuous-outcome dataset. The chosen GAMLSS framework offers greater flexibility, interpretability, and computational feasibility for the aims of our study.

Results, Figure 4 – please consider overlaying the raw data points on these curves to show how well they fit.

Response In order for us to address your comments accurately, could you please clarify what you mean by 'raw data points'? Are you referring to the underlying, unprocessed data, or to a specific subset that you believe should be considered?

Additionally, we would like to emphasize that the reported correlations are model-based estimates that consider a variety of factors, including the age of the children. These correlations are derived from a model that considers factors beyond age, including malaria incidence, precipitation, and the asset index. This provides a more comprehensive view. We present these estimates for a selection of locations to demonstrate how various factors influence outcomes.

As part of a robustness check, we estimated the pairwise correlations based solely on the model's inclusion of child age as a predictor for several countries (see Figures 1 and 2). After comparing these with the observed empirical correlations for the same countries and finding them to be very similar, we are confident that our model-based estimates accurately reflect the true relationships.

If we have misunderstood your request, please clarify. We are happy to address any points you raise and provide further explanations, if necessary.

Results, Figure 5 – not sure of the utility of this, especially from policy/decision maker perspective. The patterns are quite similar and age is proxy for other underlying proximal risk factors.

Fig. 1: Estimated pairwise correlation stratified by age (bi-monthly age bins) on data for Angola. Additionally, the empirical correlations based on bi-monthly age bins, along with their respective confidence intervals, are shown.

Fig. 2: Estimated pairwise correlation stratified by age on data for Nigeria. Additionally, the empirical correlations based on bi-monthly age bins, along with their respective confidence intervals, are shown.

Response We would like to clarify that we consider *Figure 5* important because it shows that the indicators of interest are not evenly distributed across age groups for a given location. The figure reveals a clear nonlinear pattern, which emphasizes the importance of designing and implementing targeted measures for the population segments with the highest prevalence. Additionally, *Figure 5* illustrates

that prevalence varies across different locations, offering valuable insights into spatial disparities.

From our perspective, this figure is important because it emphasizes three key points: (i) frequently observed nonlinear patterns; (ii) variation in prevalence across different areas; and (iii) how these factors can help identify the most suitable target population for interventions. *Figure 5* of the manuscript also demonstrates the effectiveness and adaptability of our chosen method, as it efficiently handles complex nonlinear relationships and spatial variation.

However, we understand the concern about how prominent the figure is in the paper. As a result, we have reduced its prominence, making it less central while ensuring that it still effectively conveys the important findings. We hope this clarifies the role of the figure in our analysis. We are happy to discuss it further if needed.

Results – it would also be useful to incorporate metrics related to country/subnational progress towards SDG 2.2 (2030). As you incorporate year in your model formulation, this could be summarised based on raw datapoints by year compared with model prediction by year.

Response Reviewer #1 raised a similar point (*Point 3 a*). We copied the response in the following paragraph:

The revised manuscript now includes the paragraph *Estimated prevalence and trends of anemia, stunting, underweight, wasting, and overweight* (L266 – 326), which provides details on the levels and trends of all five indicators within the included countries in SSA.

Additionally, we would like to clarify that individual prevalence estimates are generally more robust when derived from univariate models because these models allow for greater flexibility in specifying distributional assumptions and spatial structures. In contrast, the incorporated multivariate model is optimized for assessing co-occurrence patterns. It does not include all spatial terms for each outcome, which may modestly influence the marginal prevalence estimates. Therefore, we restrict our discussion of prevalence results to the country level and provide them primarily for contextualizing the co-occurrence analysis.

Discussion – more detailed and specific recommendations/implications of the research should be incorporated. This should also include key recommendations by country/sub-national within country.

Response We thank Reviewer #3, Reviewer #1, and Reviewer #2 – who highlighted similar aspects – for raising this important point. We now provide a literature synthesis in the *Discussion* (L433 – L448), focusing on the following aspects: (i) put our results into perspective, and compare our results (prevalence and co-occurrence metrics) to existing results; (ii) interpret the age-specific and spatial correlation patterns in light of existing studies; and (iii) better articulate the novel scientific contributions to the empirical strand of literature.

The revised *Discussion* now explains how the results, especially the high-resolution risk maps, can provide actionable guidance for ministries and implementing partners. Although the work is not causal in nature, the results directly influence the approach to achieving *SDG 2*. Specifically, this is achieved through the following points mentioned in the discussion (L449 – L477): (i) *Sharper geographic targeting of scarce resources*; (ii) *Bundling interventions*; (iii) *Addressing the age gradient*; (iv) *Continuous updating*; (v) *Early warning-system integration*.

We would like to thank Reviewer #3 for the positive review and detailed, constructive feedback provided during the first round of reviews. The comments of Reviewer #3 were instrumental in improving the quality of the manuscript, and we hope we adequately addressed all points raised by Reviewer #3 and the other two anonymous reviewers.

References

- [1] Rigby, R. A. & Stasinopoulos, D. M. Generalized additive models for location, scale and shape. *Journal of the Royal Statistical Society: Series C (Applied Statistics)* **54**, 507–554 (2005). doi:[10.1111/j.1467-9876.2005.00510.x](https://doi.org/10.1111/j.1467-9876.2005.00510.x).
- [2] Black, R. E. *et al.* Maternal and child undernutrition: Global and regional exposures and health consequences. *Lancet* **371**, 243–260 (2008). doi:[10.1016/S0140-6736\(07\)61690-0](https://doi.org/10.1016/S0140-6736(07)61690-0).
- [3] Black, R. E. *et al.* Maternal and child undernutrition and overweight in low-income and middle-income countries. *Lancet* **382**, 427–451 (2013). doi:[10.1016/S0140-6736\(13\)60937-X](https://doi.org/10.1016/S0140-6736(13)60937-X).
- [4] Madewell, Z. J. *et al.* Contribution of malnutrition to infant and child deaths in Sub-Saharan Africa and South Asia. *BMJ Global Health* **9**, e017262 (2024). doi:[10.1136/bmjgh-2024-017262](https://doi.org/10.1136/bmjgh-2024-017262).
- [5] World Health Organization (WHO). Malnutrition (2024). Retrieved April 8, 2025, from <https://www.who.int/news-room/fact-sheets/detail/malnutrition>.
- [6] Mertens, A. *et al.* Child wasting and concurrent stunting in low- and middle-income countries. *Nature* **621**, 558–567 (2023). doi:[10.1038/s41586-023-06480-z](https://doi.org/10.1038/s41586-023-06480-z).
- [7] Sturgeon, J. P. *et al.* Risk factors for inpatient mortality among children with severe acute malnutrition in Zimbabwe and Zambia. *European Journal of Clinical Nutrition* **77**, 895–904 (2023). doi:[10.1038/s41430-023-01320-9](https://doi.org/10.1038/s41430-023-01320-9).
- [8] United Nations Inter-agency Group for Child Mortality Estimation (UNIGME). Levels & trends in child mortality: Report 2024 – Estimates developed by the United Nations Inter-agency Group for Child Mortality Estimation (2025). Retrieved April 11, 2025, from <https://data.unicef.org/resources/levels-and-trends-in-child-mortality-2024/>.
- [9] Golding, N. *et al.* Mapping under-5 and neonatal mortality in Africa, 2000-15: A baseline analysis for the Sustainable Development Goals. *Lancet* **390**, 2171–2182 (2017). doi:[10.1016/S0140-6736\(17\)31758-0](https://doi.org/10.1016/S0140-6736(17)31758-0).
- [10] Osgood-Zimmerman, A. *et al.* Mapping child growth failure in Africa between 2000 and 2015. *Nature* **555**, 41–47 (2018). doi:[10.1038/nature25760](https://doi.org/10.1038/nature25760).
- [11] Burstein, R. *et al.* Mapping 123 million neonatal, infant and child deaths between 2000 and 2017. *Nature* **574**, 353–358 (2019). doi:[10.1038/s41586-019-1545-0](https://doi.org/10.1038/s41586-019-1545-0).
- [12] Kinyoki, D. K. *et al.* Mapping child growth failure across low- and middle-income countries. *Nature* **577**, 231–234 (2020). doi:[10.1038/s41586-019-1878-8](https://doi.org/10.1038/s41586-019-1878-8).
- [13] Seiler, J., Wetscher, M., Harttgen, K., Utzinger, J. & Umlauf, N. High-resolution spatial prediction of anemia risk among children aged 6 to 59 months in low- and middle-income countries. *Communications Medicine* **5**, 57 (2025). doi:[10.1038/s43856-025-00765-2](https://doi.org/10.1038/s43856-025-00765-2).
- [14] Lang, S., Umlauf, N., Wechselberger, P., Harttgen, K. & Kneib, T. Multilevel structured additive regression. *Statistics and Computing* **24**, 223–238 (2014). doi:[10.1007/s11222-012-9366-0](https://doi.org/10.1007/s11222-012-9366-0).

- [15] Klein, N., Kneib, T., Klasen, S. & Lang, S. Bayesian structured additive distributional regression for multivariate responses. *Journal of the Royal Statistical Society: Series C (Applied Statistics)* **64**, 569–591 (2015). doi:[10.1111/rssc.12090](https://doi.org/10.1111/rssc.12090).
- [16] Köhler, M., Umlauf, N. & Greven, S. Nonlinear association structures in flexible Bayesian additive joint models. *Statistics in Medicine* **37**, 4771–4788 (2018). doi:[10.1002/sim.7967](https://doi.org/10.1002/sim.7967).
- [17] Umlauf, N. & Kneib, T. A primer on Bayesian distributional regression. *Statistical Modelling* **18**, 219–247 (2018). doi:[10.1177/1471082X18759140](https://doi.org/10.1177/1471082X18759140).
- [18] Umlauf, N., Klein, N., Simon, T. & Zeileis, A. **bamlss**: A Lego toolbox for flexible Bayesian regression (and beyond). *Journal of Statistical Software* **100**, 1–53 (2021). doi:[10.18637/jss.v100.i04](https://doi.org/10.18637/jss.v100.i04).
- [19] Frévent, C. *et al.* The R package **HDSpatialScan** for the detection of clusters of multivariate and functional data using spatial scan statistics. *The R Journal* **14**, 95–120 (2022). doi:[10.32614/RJ-2022-045](https://doi.org/10.32614/RJ-2022-045).
- [20] Frévent, C. *et al.* **HDSpatialScan: Multivariate and Functional Spatial Scan Statistics** (2025). R package version 1.0.5; <https://CRAN.R-project.org/package=HDSpatialScan>.

Response to the *Comments Raised by the Reviewers* for the article:
*Co-occurrence patterns of malnutrition indicators among children in
sub-Saharan Africa (COMMSMED-24-1333A)*
submitted to
Communications Medicine

December 30, 2025

Comments Reviewer #1 (expertise: nutrition and dietetics)

Remarks to the authors:

I appreciate the thorough revisions. I am satisfied with the corrections made. The authors have addressed my comments appropriately, and the revised manuscript meets my expectations. The paper is will be of interest to the field.

Response We would like to thank Reviewer #1 for the positive and encouraging feedback. We are pleased to hear that the revisions adequately addressed the reviewer's comments and that the revised manuscript now meets the reviewer's expectations. We also appreciate the assessment that the paper will be of interest to the field.

Comments Reviewer #3 (expertise: epidemiology and biostatistics)

Remarks to the authors:

I would like to commend the authors on their very thorough, detailed, and considered responses to my comments - excellent work.

Response We would like to thank Reviewer #3 for the very positive feedback. We appreciate the acknowledgement of the thorough, detailed, and carefully considered responses to the reviewer's comments.

Discretionary remark:

Regarding my previous comment on Figure 4 (about overlaying raw data points to show fit), I fully understand that multivariable adjustment can alter the relationship between the modelled age-specific fractions and the underlying raw survey data. It may nonetheless be helpful to overlay the observed survey fractions by age and location as a broad reference – these could be dots for underlying survey fraction relating to each shaded portion. This would provide readers with a visual check that the modelled fractions are, at least approximately, consistent with the underlying data.

Response We thank the reviewer for this thoughtful suggestion and for acknowledging the distinction between multivariate model outputs and the underlying raw survey data. We agree that overlaying observed data can, in principle, be helpful as a broad visual assessment of model performance. However, in our setting this is neither feasible nor particularly informative given the high spatio-temporal resolution of the model and the large number of covariates included. This is the case at the spatio-temporal resolution (20×20 km) used in Figure 5 (Figure 4 in the initial submission). At this level of aggregation, many grid cells contain no or only very few observations, leading to highly unstable or undefined empirical age-specific fractions.

We have therefore assessed visual model performance using an alternative and more appropriate strategy, shown in Figures 1 and 2. These figures compare country-level empirical pairwise correlations, computed from the observed data, against corresponding model-based correlations obtained from a simplified specification that includes only age as covariate. The close observed alignment provides reassurance that the modeling framework captures the key age-related dependence structure present in the data.

By contrast, the full model underlying Figure 5 incorporates multiple covariates, and the resulting adjustment for multiple covariates substantially alters marginal age-specific relationships. As a result, overlaying raw empirical correlations in that illustration would not allow for a meaningful visual comparison and could be misleading. We therefore believe that the validation figures provided here are more informative and statistically appropriate for assessing model consistency with the observed data.

Fig. 1: Estimated pairwise correlation stratified by age (bi-monthly age bins) on data for Angola. Additionally, the empirical correlations based on bi-monthly age bins, along with their respective confidence intervals, are shown.

Fig. 2: Estimated pairwise correlation stratified by age on data for Nigeria. Additionally, the empirical correlations based on bi-monthly age bins, along with their respective confidence intervals, are shown.